# Changes in the Patterns and Characteristics of Youth ENDS Use over Time

**DOI:** 10.3390/ijerph19138120

**Published:** 2022-07-01

**Authors:** Stephen R. Shamblen, Melissa H. Abadi, Kirsten T. Thompson, Grisel García-Ramírez, Bonnie O. Richard

**Affiliations:** 1Pacific Institute for Research and Evaluation, One Riverfront Plaza, Suite 2100, 401 West Main Street, Louisville, KY 40202, USA; mabadi@pire.org (M.H.A.); kthompson@pire.org (K.T.T.); brichard@pire.org (B.O.R.); 2Pacific Institute for Research and Evaluation, Berkeley, CA 94704, USA; ggarcia-ramirez@prev.org

**Keywords:** ENDS, e-cigarettes, tobacco, ecological momentary assessment, youth

## Abstract

Research on youth use of electronic nicotine delivery systems (ENDS) has explored the correlates of initiation and use; however, little is known about the factors that predict continued youth use of ENDS. We used an ecological momentary assessment (EMA) burst design to explore both daily variability within adolescents over a two-week observation period and variability over time two years later (2018 and 2020). The contribution of device characteristics, motivations for use, contextual factors, and community factors to daily use occasions were explored. Youth participants (*n* = 35) at the start of the study were past two-week nicotine vapers, 14 to 17 years old, who resided within 100 miles of Louisville, KY, and reported past two-week ENDS use. Close to a quarter of participants ceased all tobacco use two years later, suggesting that some youth, despite prior regular vaping habits, may have only been experimenting with ENDS. The regular continued use of ENDS was predicted by trying to quit using cigarettes, appealing flavors, and being in locations where cigarette use was prohibited. Except for flavors, these factors did not affect ENDS use in year one. These findings suggest that tobacco policy might target ENDS use by prohibiting all tobacco use, including ENDS, in locations where smoking is already banned.

## 1. Introduction

Little is known about the patterns of electronic nicotine delivery systems (ENDS) use among youth relative to adults, despite youth being four times more likely than adults to use ENDS [1,2]. Even less is known about how youth ENDS use changes over time and the characteristics and contexts that may influence the transitions from youthful experimentation with substances to a persistent habit [3]. Yet, this research is critical, as ENDS use significantly increases the likelihood that youth will use more harmful and addictive tobacco products [4], making successful cessation less likely [5]. In addition, as adolescents transition to adulthood, they experience changes in their daily life that likely affect their opportunities and motivations for ENDS use, including living independently, greater work responsibilities, leaving school, and starting one’s own family [6]; the maturation of cognitive and emotional abilities [7]; and changes in the levels of peer and parent influence on behavior [8]. Understanding the factors that might be driving long-term youth ENDS use is important for informing comprehensive policy and prevention efforts aimed at reducing the public health impact of ENDS. As such, we sought to examine the characteristics associated with both the short-term and long-term use of ENDS.

Several characteristics have been found to be associated with youth ENDS use when measured at the same point in time, including *device characteristics*, such as flavors [9] and POD systems like JUUL [10]; *motivations for use*, such as positive ENDS attitudes and expectancies [11]; *contexts* surrounding use occasions, such as when cigarette use is prohibited [12] and parent and peer use [11]; and *community* factors, such as marketing normalizing ENDS use [13] or exposure to others using ENDS [14]. What remains unclear is whether these characteristics of youth ENDS use change over time, as well as whether they differentially predict the levels of use over time.

Longitudinal studies provide some guidance on these potential relationships, suggesting that changes in youth ENDS use occur largely due to contextual and community factors.. Specifically, family members using ENDS [15] and exposure to advertising makes the later initiation of ENDS use more likely [16,17]. However, these studies do not speak to what factors are responsible for the changes in ENDS use among those who are *already* ENDS users, limiting our understanding of the factors that may influence the trajectories of ENDS use among youth.

Similar to longitudinal studies, ecological momentary assessment (EMA) approaches have identified important factors that precede or predict youth ENDS use; however, EMA studies are usually conducted over a shorter period of time with multiple time-intensive observations. These approaches often yield more accurate measurements of ENDS use [18,19] as well as allow for more detailed, real-time information about the time-varying characteristics of youth ENDS use [20,21]. As such, EMA studies can yield uniquely helpful information for prevention and policy. Evidence from EMA studies also suggests that youth ENDS use is affected by contextual and community factors, as youth ENDS use largely occurs in environments where combustible cigarette (CC) use cannot be concealed and in social contexts with peers; however, there has been no evidence to suggest motivations (e.g., easy to get, curiosity) as a primary driver of use [21,22].

The present study used an EMA burst design [23] that accounted for variability between two two-week (daily) observation periods separated by two years. We ran two different substantive models examining (1) the changes in ENDS use behaviors and characteristics over the two years and (2) the differential prediction of greater daily ENDS use occasions across the two years. Stated differently, these models examined changes in the characteristics surrounding ENDS use over the two-year period and the predictors of more daily ENDS use occasions. Based on the literature, we expected that the predictors of ENDS use would differ over time as adolescents transitioned into young adulthood. However, given the lack of research in this area, we did not have hypotheses related to *which* factors would change over time. The literature guided the selection of the ENDS use characteristics we measured, which included: (a) device characteristics (e.g., flavors, JUUL brand), (b) motivations for use (e.g., friends use, easy to use), (c) contextual factors (e.g., who with and location), and (d) community factors (e.g., exposure to use in community, exposure to ads normalizing use). To our knowledge, no other EMA or longitudinal studies have included such a wide range of factors to investigate the influence of youth ENDS use over time.

## 2. Material and Methods

### 2.1. Participants

Data were collected as part of a follow-up to a parent project on youth vaping, which used EMA methods to examine the daily exclusive and dual use of e-cigarettes and tobacco cigarettes among youth. Participants completed all EMA survey measures on smartphones. Our sample eligibility for the follow-up study was limited to the 50 youth who participated in the first study. Eligibility criteria for the first study included youth between 13 and 17 years of age who lived within 100 miles of Louisville, KY and reported vaping nicotine in the past two weeks on a screener. Additional details on the sample and methods are published elsewhere [20,21]. Of these 50 youth, 35 participated in the follow-up study. Vaping status did not affect eligibility in the follow-up study; thus, the sample of 35 included both current and past ENDS users. Both years offered USD 15 for completing the initial online survey, USD 5 for each completed EMA survey, a USD 20 bonus if they completed all EMA surveys, and USD 15 (year one only) for returning the study phone. Due to COVID-19, participants were not offered study phones in year three and were unable to participate if they did not have personal access to a smartphone.

Our year three data collection was unplanned at year one but arose out of a fortuitous opportunity to use organizational funds to collect follow-up data over a brief three-month period. Of the year one participants, 42 (84%) expressed interest and 35 individuals completed the study (70%). Using a logistic regression, selectivity biases due to year three attrition were examined by regressing attrition status on sex, age, white race, spending money per week, and days using CC and ENDS in the 30 days prior to year one data collection. There was no evidence to suggest that these background characteristics predicted attrition (*χ*^2^(6) = 8.74, *p* = 0.19), and there was no evidence to suggest that any individual predictors significantly (*p* < 0.05) predicted attrition.

### 2.2. Procedures

#### 2.2.1. Year One Data Collection

An initial survey and two-week EMA study were conducted with a sample of 50 youth (ages 14–17) who were past two-week ENDS users in year one of data collection. The initial survey captured demographics, ENDS and combustible cigarette (CC) use in the past 30 days, and the context of ENDS use (the who, what, when, where, and why of use). Twenty-four daily surveys were collected using EMA methods for two weeks, where more fixed-time EMA observations occurred surrounding weekends. Nonetheless, the data were aggregated to daily measures to equally weight the days in our analysis and to allow comparability with the year three data. Considering only the 35 youth who participated in both years, on average, EMA data were provided on 13.51 of the 14 days per participant, resulting in 473 of 490 possible observations (97% completion rate). Data were collected between January and October of 2018. A more detailed description of these procedures appears elsewhere [24].

#### 2.2.2. Year Three Data Collection

All 50 former year one study participants were recruited through email and text invitations to participate in a follow-up study that involved another initial survey and daily surveys over 14 days that captured the same measures as in year one. The participants were informed that users and non-users were eligible. Fourteen daily surveys were collected over two weeks and used a reference point of 4 p.m. on the previous day to 4 p.m. that day for survey responses. On average, EMA data were provided on 12.71 of the 14 days, resulting in 445 of 490 possible observations (91% completion rate). Of note, year three data collection occurred early during the COVID-19 pandemic (June–July 2020), where quasi-experimental decreases were found in the availability of vaping devices for youth (by 10%) and the use of ENDS (by 7%) [25].

### 2.3. Measures

#### 2.3.1. Initial Survey

All participants completed the initial surveys in years one and three. They were asked their age, sex, and race/ethnicity. For socioeconomic status, youth were asked how much money they had in a typical week to spend on whatever they wanted, with nine response options (e.g., <USD 5; USD 5–USD 10), coded as category midpoints to approximate an interval measure (e.g., USD 5–USD 10 was coded as USD 7.50). Participants were also asked about ENDS (“vape with nicotine”) and tobacco cigarette (“smoke part or all of a tobacco cigarette”) use in the past 30 days [26].

#### 2.3.2. EMA Surveys

EMA measures, administered at both years one and three, were developed to capture daily ENDS and tobacco use behaviors as well as the potential influences of the same-day characteristics of ENDS use behaviors.

##### Tobacco Cigarettes and ENDS Use

Participants were asked to report whether they vaped nicotine during each period or day. If they said yes, they were asked how many occasions they vaped nicotine and the maximum number of puffs during each period. When there were multiple measures on weekends in year one, the total number of use occasions per day was defined as the sum of use occasions and the maximum reported puffs across all reports that day. Participants were also asked about their CC use occasions (“How many tobacco cigarettes did you smoke…”) during each period.

##### Device Characteristics, Motivations, Contextual Factors, and Community Factors Surrounding ENDS Use

When participants reported using ENDS with nicotine, they were asked more detailed questions about their *last* use occasion. Specifically, they were asked: (a) with whom they used ENDS (myself, friend, several friends, sibling, or parent); (b) what type of ENDS flavors were used (any flavor; menthol; candy, sweets, or chocolate; tobacco; mint; or fruit); (c) what characteristics their device had (e.g., whether it was their own, a JUUL device, and had a nicotine strength of 18 mg or greater); (d) where they used (someone else’s home, my home, outdoors, school, or work); and (e) a checklist of motivations for why they may have used (curiosity, a friend wanted to, saw an advertisement, trying to quit smoking cigarettes, e-cigarettes do not have an odor, it was easy to get, only e-cigarettes are allowed, it comes in flavors I like, it feels good, like the attention, like doing vape tricks, and was bored). To maintain consistency between single and multiple EMA days in year one, we used the last occasion reported for year one weekends [20,21].

### 2.4. Analysis

All inferential models were performed using random intercept regressions with binary logistic models for dichotomous and Gaussian models for continuous dependent measures using the R framework for statistical computing [27] with the *lme4* [28] and *lmerTest* [29] libraries. We first examined the level of variability between participants by calculating intraclass correlation (ICC) coefficients treating all observations as being nested within persons. Single degree of freedom likelihood-ratio tests were performed to examine whether additional variability was explained by modeling this source of variability.

We ran two different substantive models: (1) examining changes in ENDS use behaviors and contexts over the two years and (2) examining the differential prediction of *greater* ENDS use occasions across the two years. The intercept was treated as a random effect to model variability within each person (and year). The first set of models regressed daily ENDS use characteristics on a contrast representing year and statistical controls. The second set of models regressed daily ENDS use occasions on the characteristics of ENDS use and statistical controls. These analyses were performed separately for each year and in a model across both years including year as a statistical control. All models statistically controlled for age, biological sex, white race, spending money per week, and day type (weekday vs. weekend). Preliminary analyses suggested that the weekday (i.e., Sunday 4 p.m. through Friday 4 p.m.) vs. weekend (Friday 4 p.m. to Sunday 4 p.m.) contrast explained much of the variability in the day of the week [21]. The effect size *r* (or *r* = [(*t^2^*)/(*t^2^* + *df*)]^5^ for Gaussian models and *r* = *z*/*N*^5^ for dichotomous models) [30] with 95% confidence intervals was calculated to examine the magnitude of the effects. Satterthwaite degrees of freedom were used for the Gaussian models.

## 3. Results

Our presentation of the results will first report participant characteristics, then present our findings on changes in ENDS use over the two-year period, which will be followed by a report on each of the ENDS use characteristic groups examined, where within each group we will report on (a) whether the characteristics of use changed over time and (b) whether these characteristics of use differentially predicted daily use occasions.

### 3.1. Participant Characteristics

The 35 participants were well balanced in terms of biological sex (40% male), they had a mean age of 16 at year one, they were predominately white, and they more than doubled their mean spending money per week between years one (USD 56.00) and three (USD 121.67) (see Table 1). The participants’ mean age of CC initiation was 15.33 years and ENDS initiation was 14.88 years.

### 3.2. Changes in ENDS and CC Use

Slightly more than one-quarter (26%) of individuals who used ENDS at the beginning of the study indicated no use of ENDS or CC in the past 30 days at year three. Past 30-day ENDS use decreased from 97% to 73% across the two years, and past 30-day CC use slightly increased from 26% to 31%. When comparing the daily EMA measures in Table 2, there was no evidence to suggest that daily ENDS use occasions changed over the two years (5.75 to 6.90); however, the maximum number of puffs per daily occasions decreased (4.34 to 2.17). Thus, those who continued to use ENDS two years later did not decrease their use occasions but did decrease their maximum number of puffs per occasion. 

### 3.3. Device Characteristics

The characteristics of vaping devices used by youth in our study changed over time, likely corresponding to policy changes and maturation. Relative to year one, in year three, youth were less likely to use JUUL pods, less likely to use mint flavors, and more likely to use their own device. When examining the differential predictors of greater daily use occasions in Table 3, few relationships emerged; however, those that did suggested that using any flavor was related to more use occasions at year three and using one’s own device was related to more use occasions at year one.

### 3.4. Motivations for ENDS Use

The motivations for ENDS use changed over the two years. In year three, youth became less likely to use ENDS due to the motivations of seeing ads, ease of use, flavors, doing vape tricks, and being bored. In contrast, the motivation to use ENDS because of tobacco use being prohibited made use more likely at year three. Considering the differential predictors of greater daily use occasions, the only significant predictor was the motivation to use ENDS because of trying to quit using cigarettes, where it predicted more use occasions at year three, but not at year one.

### 3.5. Contextual Factors Surrounding ENDS Use

Over the course of the two years, youth who continued to use ENDS were more likely to use by themselves and less likely to use with several friends. There was no evidence to suggest that ENDS use locations changed over the two years, with use in the home being most common. Examining these variables as differential predictors of daily use occasions, youth had more use occasions when it was normative in their family at year one, reporting more use occasions when they used with their siblings, their parents, or by themselves; however, there was no evidence to suggest these relationships at year three. For those who continued to use at year three, there were fewer use occasions associated with locations outside of the home, including being at someone else’s home, being outdoors, and being at a restaurant or bar.

### 3.6. Community Factors Surrounding ENDS Use

Consistent with youth becoming less likely to use with several friends and ads becoming less of a motive for use in year three, youth were less likely to report being exposed to peers using ENDS and less likely to report being exposed to multiple sources of advertising. Similarly, exposure to multiple sources of advertising led to fewer daily use occasions in year three, but not year one.

## 4. Discussion

While all youth were past two-week nicotine ENDS users in year one, around one-quarter were no longer using ENDS or CC two years later, suggesting that ENDS use was youthful experimentation for some. However, of concern, ENDS use remained constant and did not decrease among those who were still using two years later. We found that device characteristics, motives, contextual factors, and community factors all played a role in whether daily ENDS use occurred, as well as the number of daily use occasions. Generally, daily ENDS use (i.e., whether use occurred at all in a day) was more strongly associated with these factors than the total number of ENDS use occasions that occurred daily. In interpreting the key findings of our paper, we will consider each of the large patterns in the data, which were (1) changes in ENDS use and being motivated to quit smoking, (2) changes in the use of flavors, and (3) changes in the contextual and community factors of use.

### 4.1. Changes in ENDS Use and Being Motivated to Quit Smoking

Consistent with the findings of Stanton and colleagues [5], we found that a number of ENDS users at year one stopped using ENDS and CC two years later. For those who were still using at year three, the motive of attempting to quit smoking predicted a larger number of daily ENDS use occasions than it did at year one, which was likely in service of replacing nicotine from cigarettes. While more daily ENDS use occasions among those motivated to quit smoking may seem like a negative outcome on the surface, research has shown that increasing ENDS use may be an effective cessation strategy among adults [31,32]; however, evidence is mixed for this relationship, and we are unaware of similar evidence for youth [33]. Further, as suggested in prior studies [21], the level of nicotine dependence may serve as a critical moderator in explaining why experimentation with ENDS leads to continued use two years later for some and the discontinuation of use for others. Although we did not measure nicotine dependence in our study, our findings do provide support for examining this issue further. Determining the factors that lead to dependence may be key in allaying the harmful effects of ENDS use.

### 4.2. Changes in the Importance of Flavors

The findings surrounding device characteristics largely highlight the importance of flavors, which have been found as the top reason for *initiating* ENDS use [34]. The appeal of flavors served as a greater motive for using ENDS in year one, but a lesser motive two years later. It is possible that for those who continued using ENDS, obtaining nicotine became the main motive, whereas the availability of flavors became less important. However, it is also possible that this finding represents changes in the availability of flavors due to the FDA’s restriction of cartridge-based flavors [35], JUUL Labs, Inc. halting the sale of its most popular flavors in response to criticism that their marketing tactics were targeting youth [36,37], and the reduced availability of tobacco products in general as a result of T21 [35]. In support of this interpretation, the use of mint flavors and the JUUL brand decreased over the two-year period. For this sample, another possible interpretation is that the changes in use patterns could have been influenced by a decrease in the availability of ENDS products for youth during the COVID-19 pandemic [25]. Nonetheless, the use of flavors continued to play an important role in reinforcing ENDS use behaviors. Specifically, while the use of flavors did not predict daily use in year three, flavors did predict greater use occasions on days when ENDS were used. Thus, flavors continued to influence youth ENDS use over time, but in different ways. It is also noteworthy that the use of one’s own device increased over the two-year period, which is likely a function of youth being older, having more spending money, and/or COVID-19 necessitating having one’s own device amid concerns about virus transmission [38].

### 4.3. Changes in the Contextual and Community Factors of Use

The changes in contexts among those who continued to use likely reflect movement from experimentation to patterns of regular tobacco product use. While ENDS use with friends was common in year one [21], it became less common in year three, and youth who continued to use became more likely to use ENDS by themselves. The social context of using with peers became less of a driver for ENDS use, while environmental context and community factors gained greater influence on youth behavior. For those using ENDS in year three, there was less use of ENDS in public locations, potentially where CC could be used instead. Contextual interviews with participants at year three confirmed instances of swapping ENDS and CC use and youth using CC at work and with “older friends” when in the early stages of CC initiation. These findings are similar to those found in studies with young adults, where they were more likely to use CC in contexts where they were allowed (e.g., at work or around others who use CC) and more likely to use ENDS in contexts where CC use was not allowed (e.g., traveling, in public spaces) [39]. Again, these levels and patterns of use could also reflect the circumstances of the first year of the pandemic when youth were less likely to be in school or socializing with large groups of friends due to state stay-at-home orders [40].

Interestingly, a greater exposure to ads normalizing ENDS use two years later was related to fewer days using ENDS and fewer daily use occasions, which may suggest that an increasing number of ads may have been considered as intrusive and elicited psychological reactance [41] in the form of less ENDS use. In contrast, other studies have suggested that greater exposure to ENDS advertising typically yields a lower perceived risk and a greater likelihood of subsequent use [42]. Our findings are not necessarily incongruent, as the present study found this relationship for daily use occasions among ENDS users, where other studies have examined the initiation and dichotomous measures of any use. The present study shows that this relationship may be more nuanced, where media exposure may foster initiation, but bombarding those who already use with ads may backfire and have the undesired effect of making use less likely. Marketing researchers have suggested that intrusive advertising may result in psychological reactance, where, to maintain their autonomy, advertising recipients may engage in a behavior that is the opposite of the advertiser’s intention [41,43]. While this is a positive unintended result, more research is needed to confirm this relationship with a larger sample of individuals.

### 4.4. Implications

As shown in other studies [5], some people use tobacco products for a short period and quit shortly thereafter (i.e., within two years); however, it is most important to understand why some continue to use regularly. The three primary drivers of continued ENDS use appear to be (a) appealing flavors, (b) the use of ENDS as a smoking cessation strategy, and (c) the use of ENDS as method to obtain nicotine when CC cannot be used. ENDS flavors often attract youth and keep experimental or short-term users interested. However, using ENDS for cessation, to replace cigarettes, or due to nicotine dependence appear to keep youth using for the longer term. Except for flavors, these factors affected ENDS use two years later, but did not affect ENDS use at year one. Efforts to curtail regular youth ENDS use will likely be effective when reducing these later predictors of regular use. Those who are attempting to stop using tobacco products are already moving to the desired outcome of less use; however, like their adult counterparts, youth will likely require a protracted period for cessation that will involve several quit attempts, with some being successful and others resulting in relapse [44]. Similarly, the T21 law and flavor restrictions served as an initial encouragement to quit or strengthened an ongoing quit attempt, as participants who had quit using ENDS often noted that these regulations reduced availability, especially the flavors that they enjoyed [45]. Nonetheless, these gains must be counterbalanced with tobacco companies positioning their ENDS products as a safer replacement for their CC products [46].

Fewer policy efforts have been directed towards prohibiting ENDS use in areas where CC use is not allowed. Evidence from the literature on combustible tobacco products suggests that location-specific tobacco bans are effective at reducing use [47] and that regulations may affect youth transitioning from short-term combustible tobacco product use to regular combustible tobacco product use [48]. Beyond the federal implementation of T21, currently only 16 states (AK, DC, HI, MA, NE, NH, NJ, NM, NY, ND, OR, PA, RI, SD, UT, VT, and WV) have laws that restrict the use of ENDS in the same locations where CC use is restricted. Implementing laws in the rest of the 34 states may positively impact the proportion of youth who transition from short-term to regular ENDS use.

### 4.5. Limitations

One challenge in interpreting our findings is that the year three data collection occurred during the first year of the pandemic, when some studies reported changes in the frequency, type, and access of nicotine use among youth and young adults [38,49,50] However, findings from a national survey indicated that while perceived barriers to the access and availability of nicotine contributed to changes in use patterns, these barriers did not result in decreased use [25]. In addition, while this timeframe invariably impacted the behavior of youth, we believe the changes observed in this particular set of measures were also a function of processes that were independent of the COVID-19 pandemic. Moreover, compliance with the COVID-19 stay-at-home orders was likely less in youth. In fact, one study showed that only half of youth reported being affected by the stay-at-home orders, ref [40] and another study showed that nearly one-third of youth reported breaking COVID-19 guidelines [51]. Additionally, while store closures may have initially impacted access, in-depth interviews with youth in our sample indicated that they found this to be only a temporary challenge that was resolved once their usual store reopened [38], which is consistent with other study findings [52]. Thus, while COVID-19 invariably affected ENDS use behaviors and contexts, we do not feel that these findings solely represent a historical artifact, but rather have implications for understanding and addressing youth ENDS use over time.

One other potential limitation is that our measurements were primarily daily measures, as opposed to more intensive event-based or random occasion EMA sampling. Nonetheless, studies have demonstrated a high level of reliability between daily EMA and random measurements [18]. Related to this point, some of our measures inquired about the “last time you vaped” to reduce survey burden. Thus, asking about the last occurrence of vaping with nicotine may have biased our study towards observations later in the day (e.g., the finding that only 14% of year one use occasions were at school). This bias is likely less pronounced in our measures of the frequency of use for the entire 24-h reporting period than for our measures of the characteristics of use for the last ENDS use occasion. Finally, our findings are primarily generalized to frequent (or past two-week) ENDS users in year one, as opposed to occasional users; however, we feel this is an important population for examining tobacco use behaviors, transitions, and quitting over time.

## 5. Conclusions

The present study provides evidence that there are indeed youth who experimentally use ENDS, even some who use frequently (past two week), and then cease all tobacco use, which was reflected in about a quarter of our sample. For those who did continue to use ENDS, one of the strongest motivators of ENDS use was a desire to quit smoking, suggesting some youth who continue to use ENDS are doing so as a smoking cessation strategy. The characteristics of the devices used suggest that federal regulations and T21 legislation have been somewhat effective; however, importantly, the flavors that youth could still access drove more use occasions for those who continued to use ENDS. Nonetheless, flavors became less of a motivator for any use over time for those who continued to use in year three. In addition, contextual and community factors differentially affected youth ENDS use over time, where ENDS use became less of a social activity with others, especially in locations where smoking was not allowed. Policy implications stemming from this work suggest that tobacco legislation banning the use of ENDS in locations where smoking is also not allowed may be effective in reducing youth ENDS use. Further, exposure to multiple sources of advertising may foster initial youth ENDS use; however, being exposed to more advertising may ultimately have the unexpected result of driving youth away from use. While these findings point to greater symptoms (or correlates) of nicotine dependence by youth who continue to use ENDS, future studies should explore how dependence differentially changes over time for those who continue and those who cease to use ENDS and other tobacco products.

## Figures and Tables

**Table 1 ijerph-19-08120-t001:** Participant (*n* = 35) characteristics (with standard deviations).

	Year 1	Year 3
Male	40%	
Age at Year 1	16.11 (0.90)	
White Race	89%	
Spending Money per Week	USD 56 (51.52)	USD 121.67 (114.06)
Age of CC Initiation	15.33 (1.85)	
Past 30-Day CC Use	26%	31%
Age of ENDS Initiation	14.88 (1.39)	
Past 30-Day ENDS Use	97% *	73%
Not Using CC or ENDS at Year 3	26%	

* One JUUL user was unaware that it contained nicotine when responding to this question.

**Table 2 ijerph-19-08120-t002:** Change over time as a function of year, weekday vs. weekend, and their interaction.

	Between		Means (SD)/Percent		Year		
	ICC	*χ*^2^(1) *	Year 1	Year 3	*z*/*t*	*p*	*r* (95% CI)
Level of Use							
Measures of Use							
Occasions	0.63	670.79	5.75 (12.08)	6.90 (12.79)	0.32	0.751	0.05 (±0.32)
Most Puffs	0.33	237.05	4.34 (6.99)	2.17 (3.26)	−3.55	0.001	−0.51 (±0.17)
Nicotine Strength	0.60	253.76	10.95 (6.59)	11.34 (6.78)	−0.04	0.968	−0.01 (±0.29)
Device Characteristics							
Use of Any Flavor							
Flavor Used Last Time	0.69	58.15	91%	85%	−0.98	0.327	−0.05 (±0.10)
Use of Specific Flavors							
Menthol	0.84	44.60	2%	15%	0.08	0.934	0.00 (±0.11)
Mint	0.70	59.40	26%	1%	−3.32	0.001	−0.19 (±0.11)
Fruit	0.45	73.43	61%	79%	1.66	0.098	0.09 (±0.11)
Candy, Sweets, or Chocolate	0.64	18.14	8%	6%	0.38	0.706	0.02 (±0.11)
Use of PODS							
JUUL Brand	0.81	170.74	64%	10%	−4.84	<0.001	−0.26 (±0.10)
Use of Own Device							
My Own Device	0.59	93.73	53%	98%	2.67	0.008	0.14 (±0.10)
Motivations for Use							
Curiosity	0.79	69.35	12%	2%	−0.66	0.508	−0.03 (±0.10)
Friend	0.62	81.04	23%	5%	−1.07	0.284	−0.06 (±0.10)
Ad	0.97	92.07	8%	0%	−5048.34	<0.001	<−10.00 (±0.09)
Quitting Cigarettes	0.97	123.28	14%	3%	−1.11	0.267	−0.06 (±0.10)
No Odor	0.78	208.79	44%	35%	−0.93	0.352	−0.05 (±0.10)
Easy	0.49	100.98	75%	46%	−2.11	0.035	−0.11 (±0.10)
Tobacco Prohibited	0.65	144.14	20%	30%	455.44	<0.001	>10.00 (±0.10)
Flavors	0.67	171.35	80%	50%	−2.84	0.004	−0.15 (±0.10)
Feels Good	0.67	82.00	85%	75%	−0.84	0.399	−0.04 (±0.10)
Attention	0.89	54.05	6%	6%	−0.58	0.565	−0.03 (±0.10)
Vape Tricks	0.68	152.48	61%	30%	−2.76	0.006	−0.14 (±0.10)
Bored	0.63	137.12	68%	42%	−2.23	0.026	−0.11 (±0.10)
Contextual Factors							
With Whom							
Myself	0.49	93.96	38%	68%	2.55	0.011	0.13 (±0.10)
Friend	0.21	28.51	22%	28%	0.58	0.559	0.03 (±0.10)
Several Friends	0.28	41.15	37%	32%	−133.45	<0.001	<−1.00 (±0.10)
Sibling	0.93	92.53	5%	13%	0.59	0.556	0.03 (±0.10)
Parent	0.94	89.45	2%	11%	0.34	0.730	0.02 (±0.10)
Where							
Someone Else’s Home	0.20	11.27	25%	21%	−0.01	0.992	0.00 (±0.10)
My Home	0.27	41.93	40%	54%	1.69	0.091	0.09 (±0.10)
Outdoors	0.40	21.88	8%	9%	−0.29	0.773	−0.01 (±0.10)
Restaurant or Bar	0.00	0.00	0%	1%	0.43	0.667	0.02 (±0.10)
School	0.65	29.83	14%	0%	-	-	-
Work	0.43	26.96	6%	15%	0.73	0.468	0.04 (±0.10)
Community Factors							
Expos. to Other’s ENDS Use							
Expos. to Adults Using ENDS	0.42	182.17	30%	25%	−1.49	0.137	−0.05 (±0.07)
Expos. to Peers Using ENDS	0.31	135.96	65%	44%	−4.72	<0.001	−0.16 (±0.07)
Expos. to Advertising							
Expos. to Three Advertising Srcs.	0.43	389.25	0.87 (1.07)	0.37 (0.76)	−2.85	0.007	−0.42 (±0.20)

* All tests significant *p* < 0.001, except for restaurant or bar, where *p* = 0.999.

**Table 3 ijerph-19-08120-t003:** Predictors of use occasions at years one, three, and combined.

	Year 1			Year 3			Combined		
	*t*	*p*	*r* (95% CI)	*t*	*p*	*r* (95% CI)	*t*	*p*	*r* (95% CI)
Device Characteristics									
Use of Any Flavor									
Flavor Used Last Time	0.93	0.354	0.07 (±0.14)	2.17	0.032	0.18 (±0.15)	2.28	0.024	0.13 (±0.11)
Use of Specific Flavors									
Menthol	0.41	0.680	0.03 (±0.16)	0.16	0.878	0.04 (±0.46)	0.70	0.486	0.04 (±0.12)
Mint	−1.79	0.076	−0.14 (±0.16)	−0.99	0.330	−0.19 (±0.39)	−2.22	0.028	−0.13 (±0.12)
Fruit	−1.44	0.153	−0.12 (±0.16)	−0.10	0.922	−0.02 (±0.48)	−1.95	0.052	−0.12 (±0.12)
Candy, Sweets, or Chocolate	−1.12	0.266	−0.09 (±0.16)	−0.18	0.856	−0.04 (±0.45)	−1.71	0.089	−0.10 (±0.12)
Use of PODS									
JUUL Brand	0.84	0.405	0.08 (±0.19)	1.08	0.282	0.10 (±0.18)	1.48	0.141	0.10 (±0.13)
Use of Own Device									
My Own Device	3.21	0.002	0.22 (±0.12)	0.49	0.625	0.04 (±0.17)	3.05	0.003	0.16 (±0.10)
Motivations for Use									
Curiosity	0.85	0.398	0.06 (±0.14)	−0.55	0.582	−0.05 (±0.17)	0.23	0.817	0.01 (±0.11)
Friend	−0.47	0.640	−0.03 (±0.14)	−1.08	0.282	−0.10 (±0.18)	−0.87	0.383	−0.05 (±0.11)
Ad	0.01	0.989	0.00 (±0.14)	-	-	-	0.10	0.917	0.01 (±0.11)
Quitting Cigarettes	0.06	0.951	0.00 (±0.14)	2.06	0.042	0.19 (±0.17)	0.62	0.537	0.03 (±0.11)
No Odor	−1.23	0.219	−0.09 (±0.15)	−0.60	0.551	−0.05 (±0.18)	−1.39	0.166	−0.08 (±0.11)
Easy	0.58	0.560	0.04 (±0.14)	1.30	0.195	0.12 (±0.17)	1.39	0.166	0.08 (±0.11)
Tobacco Prohibited	0.00	0.997	0.00 (±0.14)	−1.20	0.234	−0.11 (±0.18)	−0.83	0.408	−0.05 (±0.11)
Flavors	1.61	0.110	0.12 (±0.14)	1.45	0.149	0.13 (±0.16)	2.17	0.031	0.12 (±0.10)
Feels Good	0.33	0.743	0.02 (±0.14)	−0.07	0.945	−0.01 (±0.17)	−0.02	0.982	0.00 (±0.11)
Attention	1.03	0.306	0.08 (±0.14)	−0.67	0.502	−0.06 (±0.19)	1.07	0.285	0.06 (±0.11)
Vape Tricks	1.21	0.228	0.09 (±0.14)	0.60	0.551	0.05 (±0.17)	1.37	0.171	0.08 (±0.11)
Bored	0.19	0.851	0.01 (±0.14)	−0.36	0.719	−0.03 (±0.18)	−0.59	0.559	−0.03 (±0.11)
Contextual Factors									
With Whom									
Myself	2.03	0.043	0.14 (±0.13)	1.52	0.131	0.13 (±0.16)	2.19	0.030	0.12 (±0.10)
Friend	0.07	0.946	0.01 (±0.14)	1.41	0.163	0.13 (±0.17)	0.59	0.554	0.03 (±0.11)
Several Friends	0.76	0.446	0.06 (±0.14)	0.43	0.669	0.04 (±0.17)	−0.07	0.947	0.00 (±0.11)
Sibling	3.51	0.001	0.25 (±0.12)	−1.90	0.059	−0.16 (±0.17)	2.51	0.013	0.14 (±0.10)
Parent	2.06	0.040	0.15 (±0.14)	0.65	0.516	0.06 (±0.16)	2.82	0.005	0.15 (±0.10)
Where									
Someone Else’s Home	0.64	0.526	0.05 (±0.14)	−2.27	0.025	−0.20 (±0.17)	−0.35	0.729	−0.02 (±0.11)
My Home	0.99	0.322	0.07 (±0.14)	−1.83	0.069	−0.16 (±0.18)	0.23	0.818	0.01 (±0.11)
Outdoors	−0.66	0.513	−0.05 (±0.14)	−3.40	0.001	−0.30 (±0.17)	−2.16	0.031	−0.12 (±0.11)
Restaurant or Bar	0.03	0.974	0.00 (±0.15)	−2.61	0.010	−0.23 (±0.17)	−1.43	0.153	−0.08 (±0.11)
School	−0.14	0.886	−0.01 (±0.14)	-	-	-	-	-	-
Work	−0.70	0.486	−0.05 (±0.15)	−1.60	0.112	−0.14 (±0.18)	−0.05	0.957	0.00 (±0.11)
Community Factors									
Exposure to Other’s ENDS Use									
Expos. to Adults Using	0.45	0.657	0.02 (±0.09)	1.33	0.184	0.07 (±0.10)	1.41	0.159	0.05 (±0.07)
Expos. to Peers Using	2.17	0.031	0.10 (±0.09)	1.96	0.051	0.10 (±0.10)	2.59	0.010	0.09 (±0.07)
Exposure to Advertising									
Expos. to Three Sources	−0.38	0.703	−0.02 (±0.09)	−2.53	0.012	−0.13 (±0.10)	−1.74	0.082	−0.06 (±0.07)

## Data Availability

The data presented in this study are available on request from the corresponding author. The data are not publicly available due to concerns with participant anonymity (i.e., minors) and a lack of sharable documentation for the full dataset.

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
