# Peer review of "Changes in the Patterns and Characteristics of Youth ENDS Use over Time"

_ijerph, 2022, doi:10.3390/ijerph19138120_

Round 1
Reviewer 1 Report
It is a very interesting study, with a well-developed methodological approach, despite the difficulties inherent in the COVID period. In any case, the findings are very relevant. Certainly, it is a manuscript with little room for improvement in my opinion.
However, I do consider it advisable for the authors to expand the literature review on the effectiveness of ENDS in smoking cessation processes. Some of the findings favorable to its use for this purpose have presented certain conflicts since they are financed by the industry. While others have clearly qualified that its effectiveness is conditioned by several elements, such as that its use is framed within a process of taboo cessation with psychological support. It is also noted that it is important the experience of the consumer in the use of these devices, the depth of the draught, and other elements that determine that its use for smoking cessation presents promising results. High rates of dual use of conventional cigarettes and ENDS are also reported. In relation to all of the above, it would be useful to clarify that the findings of the previous literature are referred to the adult population, and therefore not applicable to the adolescent population.
Author Response
Thank you for your kind words. We agree that (a) there is conflicting evidence about whether tobacco cessation attempts are aided by ENDS use and (b) it is unclear whether findings for young adults apply to youth. We have revised the sentence to read, “While more daily ENDS use occasions among those motivated to quit smoking may seem like a negative outcome on the surface, research has shown that increasing ENDS use may be an effective cessation strategy among adults, but evidence is mixed for this relationship and we are unaware of similar evidence for youth.”
Reviewer 2 Report
CENTRAL AND GENERAL ISSUES
Summary
This paper analyzes youth electronic nicotine delivery systems (ENDS) use. The findings suggest that tobacco policy could target ENDS use by preventing ENDS use in places where cigarette use is normally allowed.. However, I think there are some important aspects that need to be improved before recommending its publication in International Journal of Environmental Research and Public Health.
Specific Comments
1. The introductory section does not explicitly talk about how tobacco companies are using electronic devices and heated tobacco to replace traditional cigarettes with these new alternatives. A recent article shows how Philip Morris International is using heated tobacco to replace the traditional cigarette. In this line, I think the authors should reflect on this and cite the following paper:
Golpe, A. A., Martín-Álvarez, J. M., Galiano, A., & Asensio, E. (2022). Effect of IQOS introduction on Philip Morris International cigarette sales in Spain: a Logarithmic Mean Divisa Index decomposition approach. Gaceta Sanitaria. https://doi.org/10.1016/j.gaceta.2021.12.007
2. In the conclusions I see that they have concluded about the findings found and have marked lines of future research. However, I do not see any paragraph showing the limitations of this work. It would be important for the limitations of this paper to be made clear.
Author Response
We have added the following sentence to address this point and have cited this article, “Nonetheless, these gains must be counterbalanced with tobacco companies positioning their ENDS products as a safer replacement for their CC products.” The two paragraphs prior to the conclusion section addresses the limitations of our study. We have noted that collecting data during the COVID-19 pandemic and not using more intensive EMA methods (i.e., daily vs. multiple random daily observations) are potential limitations of our study. Please let us know if there were specific limitations that we missed in our discussion.
Reviewer 3 Report
I consider that the subject of the manuscript is very interesting. ENDS’ popularity has grown over the years and has become a public health concern. Young people are especially exposed, and it is important to understand the factors that influence the short-term and long-term ENDS use patterns.
The research presented in the present paper is well organized. The Materials and Methods section describes clearly and in detail the main points of the study: participants selection, data collection and data analysis.
The Results and Discussions sections are easy to follow, and I consider that the authors used an adequate number of references to support and compare their findings.
In conclusion, I have only one observation regarding the manuscript: the references (both in the text and in the final list) are not written according to the Instructions for Authors.
Author Response
Thank you for your kind words. We believe the references are in the appropriate format in the current production draft.